# (In)Security of File Uploads in Node.js

## ABSTRACT

File upload is a critical feature incorporated by a myriad of web applications in an effort to enable users to share and manage their files conveniently. It has been used in many useful services such as file-sharing and social media. While file upload is an essential component of web applications, the lack of rigorous checks on the file name, type, and content of the uploaded files can result in security issues, often referred to as Unrestricted File Upload (UFU). In this study, we analyze the (in)security of popular file upload libraries and real-world applications in the Node.js ecosystem. To automate our analysis, we propose and implement NODESEC– a tool designed to analyze file upload insecurities in Node.js applications and libraries. NODESEC generates unique payloads and thoroughly evaluates the application's file upload security against 13 distinct UFU-type attacks. Utilizing NODESEC, we analyze the most popular file upload libraries and real-world applications in the Node.js ecosystem. Our analysis results reveal that some real-world web applications are vulnerable to UFU attacks and disclose serious security bugs in file upload libraries. As of this writing, we received 19 CVEs and two US-CERT cases for the security issues that we reported. Our findings provide strong evidence that dynamic features of Node.js applications introduce security shortcomings and that web developers should be cautious when implementing file upload features in their applications. Finally, combining our responsible disclosure experience and root cause analysis, we identified the main causes of significant security weaknesses in file uploads in Node.js.

## CCS CONCEPTS

• **Security and privacy** → **Web application security**.

## KEYWORDS

Web Security, Node.js, Unrestricted File Upload

**ACM Reference Format:**
Anonymous Author(s). 2024. (In)Security of File Uploads in Node.js. In *Proceedings of In Proceedings of the ACM Web Conference 2024 ('WWW).* ACM, New York, NY, USA, 14 pages. https://doi.org/XXXXXXX.XXXXXXX

## 1 INTRODUCTION

File upload is a critical feature incorporated by a myriad of web applications to share and manage their files conveniently. It has been used in many useful services such as file-sharing and social media. While file upload is an essential part of web applications, it

'WWW, May 13–17, 2024, Singapore
© 2024 Association for Computing Machinery.
ACM ISBN 978-x-xxxx-xxxx-x/YY/MM. . . $15.00
https://doi.org/XXXXXXX.XXXXXXX

also extends the attack surface in these applications by creating an opportunity for adversaries to upload malicious payloads to web applications. In particular, the lack of rigorous checks on the file name, type, and content of the uploaded files can result in security issues, often referred to as the Unrestricted File Upload (UFU) [1]. A successfully uploaded malicious payload can cause potential code execution on both the client side or the server side of the web application.

The popularity of Node.js has grown significantly over the years with over one billion downloads [1]. It is especially preferred by major companies to develop scalable high-traffic web applications [2]. In Node.js, third-party library usage has become a *de facto* standard among Node.js developers. In this manner, Node.js developers frequently depend on third-party libraries to important features such as file upload, authentication, logging, etc. that require security attention [3]. Given the sheer number of existing security issues and implementation mistakes discovered in the third-party libraries [4, 5], we argue that there is a dire need to further investigate the security of the file upload libraries in Node.js.

While previous work proposed tools for identifying UFU vulnerabilities in other ecosystems [6–8], there is a dire need for a tool that is tailored for testing Node.js applications. As existing tools face challenges when applied to Node.js applications due to their unique features such as syntax, file handling, and distinct execution environments. Consequently, adapting these existing tools to test Node.js applications requires significant domain expertise and in-depth knowledge of the existing tools' source code. Moreover, the techniques employed by other tools primarily focus on identifying the incompleteness of the file upload checks in web applications rather than examining potential implementation mistakes in security-related functions. Considering almost 93.2% of the code in Node.js applications comes from third-party libraries [3], a tool tailored for Node.js applications must also be capable of detecting implementation issues in the file upload libraries.

Motivated by the urgent need, in this work, we propose NODESEC to analyze Node.js applications and file upload libraries against UFU attacks. Our tool includes 13 distinct UFU-type attacks, derived from a thorough review of previously published UFU-related CVEs from 2002 to February 2023 OWASP resources [1, 9], and GitHub issues [10, 11]. Furthermore, we conducted an exhaustive literature review by examining UFU vulnerabilities across various ecosystems and adapted these attacks to the Node.js environment, if applicable. Finally, we examined common security mistakes made by developers [12] and previous bugs [10] to identify possible implementation errors specific to Node.js developers. This methodology ensures that NODESEC includes a broader range of attack vectors than existing literature and is capable of identifying UFU-related security issues in real-world web applications and security flaws in the file upload libraries within the Node.js ecosystem.

Based on our investigation, we identified three generic objectives that a secure file upload implementation must meet to prevent

---

[1] https://nodejs.org/metrics/

UFU attacks. We leverage NodeSec to analyze the six most popular server-side file upload libraries, which, on average, have two million weekly downloads. Our findings revealed that none of these libraries fulfilled all three objectives, as they contained implementation mistakes in their validation functions and lacked critical security measures, potentially exposing them to attack vectors that allow the uploading of malicious payloads that would affect millions of live web applications using these libraries to implement their file upload feature. Moreover, we examined 11 popular real-world web applications written in Node.js using NodeSec. Our analysis revealed that these applications are not resilient against UFU due to several reasons: 1) they overrely on file upload libraries for security 2) they make errors when configuring security options in the libraries, and 3) they fail to address all edge cases in their custom implementation. Our analysis also revealed that some of these real-world web applications use additional packages or custom implementations to prevent UFU attacks. Our findings clearly demonstrate that web application developers should not blindly trust a file upload library despite its popularity in the ecosystem and instead should implement their own metadata and content checks or use additional packages that implement these security checks for uploaded files. The demonstration of NodeSec and demo videos of our findings are available at https://sites.google.com/view/wwwpaper/home

**Contributions.** Our contributions are as follows:

- **Comprehensive Analysis of UFU in Node.js:** For the first time in the literature, we investigate the security posture of the Node.js file upload ecosystem. We outline three objectives for securely handling the file upload process in Node.js applications and present prevention methods and mechanisms, that can be used by web application developers to prevent file upload attacks and satisfy objectives.
- **Node.js UFU Analysis Tool:** We implement and open source [2] NodeSec– a tool designed to analyze file upload insecurities in Node.js applications and file upload libraries. This tool will serve as a valuable resource for the community to test their web applications and libraries against file upload insecurities before deployment.
- **Evaluation of Web Applications & Libraries:** Utilizing NodeSec, we investigate the security of the Node.js file upload ecosystem. Our analysis discloses that some real-world web applications are insecure against our attacks and we disclose serious security bugs in the popular file upload libraries. As of this writing, we received 19 CVEs and two US-CERT cases for the security issues that we reported.
- **Root Causes & Recommendations:** Based on our knowledge gained from these experiments and experience from responsibly disclosing these security issues to developers we enumerate root causes leading to the insecurity of the Node.js file upload ecosystem and our recommendations.

**Responsible Disclosure.** We responsibly reported the security issues we found in all of the server-side file upload libraries and real-world web applications, resulting in 19 CVEs at the time of this writing. We also notified the US-CERT about the issues and they acknowledged our findings. We detail our vulnerability disclosure process in Section 8 in the Appendix.

---

[2]We will share the link upon the acceptance of the paper.

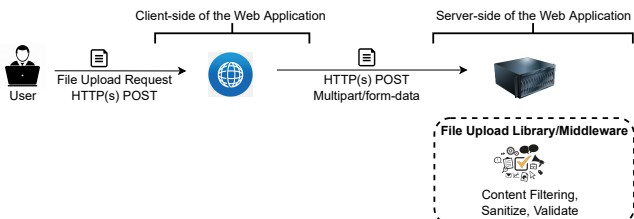

**Figure 1: A typical scenario of file upload procedure in web applications.**

## 2 UNRESTRICTED FILE UPLOAD

In this section, we discuss the general file upload process and unrestricted file upload in Node.js applications.

### 2.1 File Upload in Node.js

A typical scenario of the file upload process with HTTP(S) protocol is shown in Figure 1. First, the user picks a file and initiates the upload process to a web application. Following that, the user's client machine sends an HTTP(S) request that contains a multipart/form-data [13] form of the uploaded file to the web application server. As shown in Figure 1, the web application can perform various checks to validate the uploaded file both on the client and the server. For instance, it can validate the file type by checking its MIME (i.e., Content) type on the server side. If the file type is not valid or is undesired, the web application may reject the uploaded file to prevent any unintended file on the web application server.

**File Upload Library Usage in Node.js Applications.** Third-party libraries have become an integral component of the Node.js ecosystem. In a typical Node.js application, over 90% of the code originates from third-party libraries [3]. To properly handle the file upload process, Node.js developers often utilize server-side file upload libraries available in the Node Package Manager (npm). For example, the file upload library formidable is downloaded over 8.6 million times weekly and it is used by 1.5 million open-source applications [14]. These libraries serve as middleware for Node.js web applications, providing file upload capabilities and implementing the necessary security checks and validations to prevent any attacks that could be exploited via uploading a file.

While using third-party libraries is convenient for developers to develop Node.js applications, it comes with security risks [5, 15]. Recent incidents [16, 17] and studies have shown that developers tend to assume that popular third-party libraries are safe to use, which can result in the introduction of vulnerabilities into applications. When a vulnerability exists in a middleware or library or it is insecure, it can transmit to a web application and potentially impact a vast number of Internet users. Conversely, If the library (i.e., middleware) handled the security checks, the web application would be automatically secured, in contrast to the situation where the checks were not done by the library. Furthermore, addressing vulnerabilities in third-party packages can be time-consuming, and developers may not prioritize fixes [5]. Consequently, this practice poses a significant security threat to the entire web ecosystem.

```
1  const checkFileIsValid = (fileData, types, extensions)=>{
2      const type = fileData.mimetype;
3      if (types.includes(type) && extensions.includes(
            fileData.ext)) {
4          return true; }
5      return false; };
```

**Listing 1: A sample code snippet from the GhostCMS implementing content-filtering checks that can lead to UFU vulnerabilities.**

## 2.2 Unrestricted File Upload in Node.js

Unrestricted File Upload is a type of security weakness that enables an attacker to upload malicious files to a web application server. Even if the target server does not immediately execute the uploaded file, the web application can still be considered vulnerable or insecure with respect to UFU. This is because the presence of malicious files in the web application server poses a natural security risk as malicious content in the file can later be executed in various ways, such as exploiting other vulnerabilities in the third-party packages, direct object references, or even tricking users/admins into executing it [1]. Once the file is executed, depending on the content of the file, this vulnerability can lead to server-side attacks such as RCE [18], or client-side attacks such as XSS [19].

Listing 1 shows a code snippet from the GhostCMS Node.js application [20] that implements security checks for uploaded files through the checkFileIsValid function. Line 3 checks if the file's MIME type and extension are included in the provided arrays. However, relying solely on MIME type and file extension for validation introduces security risks, as both can be easily bypassed by an attacker [21]. Consequently, the web application remains susceptible to file upload vulnerabilities, potentially leading to arbitrary code execution on either the client or server side. For example, an attacker could upload a malicious client-side script (e.g., HTML) that executes in the victim's browser context when the file's URL is accessed. The primary goal of the attacker is to exploit the UFU security weaknesses within the target Node.js application to execute malicious code on the client-side or server-side that can lead to various types of attacks.

The consequences of UFU attacks may differ depending on the specific features and functionalities of the programming language of the server. For instance, languages like PHP can directly execute code embedded in uploaded files if they are placed in an executable path and have the executable file extension (e.g., .php) [22]. In contrast, as Node.js applications do not directly map URLs to file paths by default, it does not automatically execute code from uploaded files in response to HTTP requests [23]. Although this reduces the likelihood of direct arbitrary code execution on the server, it does not completely eliminate the possibility of UFU vulnerabilities, as the presence of a malicious file in the web application server introduces an inherent security weakness to the web application. As evading security checks in a web application to upload a malicious payload is itself unintended and exposes the application's insecurity and attackers can still exploit UFU vulnerabilities in various ways as explained in Section 11 in Appendix.

## 3 METHODOLOGY

In this section, we outline our methodology for analyzing UFU on file upload libraries and Node.js applications.

### 3.1 Attack Identification

Our research includes the following objectives: 1) identifying techniques that adversaries can employ to exploit UFU 2) investigating the common mistakes made by developers on file upload libraries 3) evaluating the effectiveness of Node.js libraries in preventing UFU vulnerabilities. We initiated our analysis by exploring the CVEs related to UFU. To achieve this, we developed a Python script that utilizes the NIST National Vulnerability Database (NVD) API, allowing us to extract all the CVEs from the NVD dataset [24]. Utilizing this script, we retrieved all CVEs ranging from the year 2002 to February 2023. This provided us with a comprehensive dataset that includes over two decades of vulnerability data, consisting of a total of 221,549 CVEs. Given the scope of this work, we checked the descriptions and Common Weakness Enumeration (CWE) codes of the CVEs. In particular, we checked for terms such as 'file upload', 'unrestricted upload', 'arbitrary file upload', and also the CWE-434 'Unrestricted Upload of Files with Dangerous Types'. Our final dataset includes 1846 CVEs that are all related to the UFU vulnerabilities. We also checked for any false positives or duplicates and removed them from the dataset. To make our dataset more comprehensive to identify our attacks we analyzed OWASP [9, 25] sources and GitHub issues of the file upload libraries [10, 11] to investigate recent techniques employed by adversaries to exploit UFU in Node.js applications. Furthermore, we examined the common security mistakes made by the developers [12] and previous GitHub issues of libraries to identify implementation errors. Lastly, we conducted an exhaustive literature review on prior research examining UFU vulnerabilities across various ecosystems [6–8], adapting these attacks to the Node.js environment, if applicable. From our analysis, we selected 13 different attacks, grouped into three categories: (1) File Name-based Attacks (File Extension Injection, Null Byte Injection, Script-Named File Name, Path Traversal, Overwrite), (2) File Type-based Attacks (Spoofing-based, Polyglot File Attacks, Executable File Attack), and (3) File Content-based Attacks (PDF File Attacks, SVG Upload Attack).

### 3.2 Attack Descriptions

In this section, we provide detailed descriptions of the attacks and their respective categories considered in this work. We elaborate on the impact of each attack, discuss its possible consequences, and explain their differences.

**File Name-based Attacks** In this attack category, an adversary modifies the file name to alter the intended logic of the web application or injects malicious characters into the file name to abuse the file upload inputs of the web application. The malicious characters can be in the form of multiple file extensions, without file extension, null bytes, scripts, or non-alphanumeric characters [26]. Further details about different file naming-based attacks that are employed by NODESEC are articulated in below.

- *[A1] File Extension Injection:* In this attack, an adversary modifies the extension of the file name to exploit improper file name extension controls in the web application. For example, an attacker

can inject multiple file extensions to bypass the file validation logic based on file extensions [27].

- *[A2] Null Byte Injection*: In this attack, the attacker inserts a null-byte into a file name to alter the intended logic of the application. An attacker can inject different portions of the file name to perform this type of attack. For example, similar to the File Extension Injection attack (A1), an attacker injects a null byte between a forbidden extension and an allowed extension to alter the intended logic of the targeted Node.js application [28].
- *[A3] Script-named file name:* In this attack, the attacker inserts a script into a file name, such as an XSS payload, which may trigger the execution of the payload in the victim's browser if the name of the uploaded file is not sanitized properly.
- *[A4] Path Traversal:* In this attack, the adversary inserts malicious characters into the file name to achieve path traversal attacks, potentially allowing them to access directories outside the restricted directory in the Node server [29].
- *[A5] Overwrite Attack*: In this attack, an adversary aims to overwrite a file on a target web application server, particularly server configuration files, to maliciously change the server settings. With this attack, an attacker can externally control critical configuration files that play a crucial role in the operation of the target web application,

***File Type-based Attacks*** In these attacks, an adversary modifies the file type of the malicious payload. In the next subsections, we explain the details of how it can be realized to create a malicious payload.

***Spoofing-based Attacks.*** In these attacks, the adversary bypasses the file type validation logic of the web application by spoofing the file content-type (MIME-type) and magic header bytes of a file [30]. These attacks can lead to the execution of malicious code on the server or client-side, unauthorized access, and data leaks [1].

- *[A6] MIME Type Spoofing:* A file's content-type represents the file's MIME type, which describes the file and its structure. File upload libraries may use MIME types to validate file types. However, an attacker can easily bypass this attempt by modifying or spoofing the content type of the file. If the target server relies only on the MIME type check to validate the file content, MIME type spoofing can enable the attacker to bypass the checks and upload a malicious payload file, potentially leading to code execution on the server-side.
- *[A7] Magic Byte Spoofing:* Another technique used to validate file types is checking the magic header byte [31]. An attacker can create a malicious file, such as a script, and change the magic byte to other file types, such as a PNG file, to bypass the file type validation checks performed by the web application. This attack can lead to the execution of malicious code on the server.

***Polyglot File Attacks.*** Polyglot files are files that are valid in multiple different file formats, allowing adversaries to create these files to hide malicious payloads and bypass the file type validation logic of the web application [32, 33]. Unlike spoofing-based attacks, where an adversary only changes the magic bytes and/or MIME type of the file, polyglot files are constructed by merging the syntax and semantics of multiple file formats [34]. As a result, a web application might be resilient against spoofing-based attacks but still be vulnerable to polyglot file attacks. Polyglot files can be used to

inject malicious scripts and bypass the content security policy of the file upload mechanism of a web application, leading to various types of attacks such as XSS and RCE.

- *[A8] JS+JPEG Polyglot:* This type of polyglot file is valid in both JPEG and JS file formats. If the content-filtering mechanism of the web application accepts it as a JPEG file, it will be uploaded to the server. Once the file is uploaded to the web application server, the attacker can execute the malicious payload by remotely accessing the file or during the parsing it can cause to server to down [33].
- *[A9] HTML+PDF Polyglot*: A PDF+HTML polyglot file is valid in both PDF and HTML file formats. It can be used by adversaries to bypass the content security checks of web applications and insert a malicious payload within the HTML file. Similar to the JS+JPEG Polyglot file, the attacker can execute the malicious payload by remotely accessing the file from the browser [35].

***[A10] Executable File Upload Attack.*** In this attack, an attacker uploads an executable file (e.g., EML, HTML) that is possible to be executed on the client or server side of a web application. In this attack, an attacker uploads an HTML payload file to a target web application. The uploaded payload file can redirect a victim to a malicious website or execute a JavaScript payload embedded on an HTML file [25].

***Content-based Attacks*** In these attacks, an adversary embeds malicious content into a seemingly benign file, such as a PDF or SVG. Although these types of attacks can be achieved by inserting malicious content into different file types. We focus on PDF and SVG files due to their popularity in the web ecosystem in general and their potential malicious impacts.

***PDF File Attacks.*** PDF is one of the most popular file formats used in web applications [36]. Web applications, such as PDF editors, may render the PDF file on the server side to display the document to the user. Additionally, web applications used by law firms (e.g., DocuSign) may require users to upload necessary information in PDF format and store it on the server side. The structure of PDF files can be abused by adversaries by embedding a JavaScript payload or compressing the content to exhaust the resources of the target server [37].

- *[A11] JavaScript Embedded PDF*: In this attack, adversaries inject malicious JavaScript code inside a PDF document. For instance, an attacker can inject a JavaScript payload into a PDF document and upload it to a web application to perform a stored XSS attack [38].
- *[A12] PDF Bomb Attack*: This attack involves adversaries abusing the encoding options of a PDF file to compress the streams. Once the malicious PDF file is uploaded to a web application server, it decompresses the content, causing resource exhaustion on the target server [36].

***[A13] SVG File Upload Attack.*** SVG file attacks exploit the features of SVG files, which support inline JavaScript code. In this attack, the adversary injects a JS payload into an SVG file to achieve different types of attacks, such as XSS [39].

### 3.3 Secure File Upload Validation Objectives

Our goal in this section is to enumerate a set of objectives that should be implemented by web applications utilizing file upload features. We enumerate these objectives by searching the following resources: secure file upload implementation principles [1, 9],

prevention techniques and tools against the attacks enumerated in Section 3.2, secure file upload implementation practices in other ecosystems.

Our search resulted in two types of techniques: 1) Techniques to validate the uploaded files, and 2) Techniques to minimize the risks of malicious file uploads. The former includes efforts to identify malicious intent from the file and methods to prevent them, while the latter includes other practices such as storing the uploaded files on a different server, authentication, and authorization mechanisms, and file size and upload request limiting mechanisms against DoS attacks. In this paper, we only study the attacks that can be performed by modifying the uploaded files, we consider the techniques in the latter as out of scope. In conclusion, a file upload mechanism in the web application should satisfy the following file upload validation objectives to prevent UFU attacks:

- *Objective-1: File Name Validation:* An adversary can abuse the file name of the uploaded file to trigger a UFU vulnerability in the web application, as given in Section 3.2. To validate the file name, a web application can assign a randomly generated safe string such as UUID to the file name [40]. Additionally, a web application can also sanitize the file name of an uploaded file by removing malicious characters before it is uploaded to the server. The developers either implement their own sanitization functions or integrate third-party packages [41, 42] to sanitize the file name.
- *Objective-2: File Type Validation:* A web application can check the MIME type of the uploaded file from the file upload request and reject the uploaded files with unexpected MIME types. While this method prevents crude file upload attacks, the MIME type of file provided by the client cannot be trusted and it can be easily spoofed by the attacker. Validating the file type directly from the content of the uploaded file is robust against such spoofing techniques. Depending on the expected file type of the web application, the techniques for the validating file type can differ [43]. A developer can make use of open-source packages by detecting the file type by its metadata [44, 45] or file stream [46].
- *Objective-3: File Content Sanitization:* An adversary can insert malicious content (i.e., script) into a seemingly benign file to trigger UFU vulnerability on the server or client side. To prevent this, a web application should sanitize the malicious content in the file. The sanitization technique can differ based on the file type [43]. A developer can utilize open source packages [47, 48] to sanitize the file or set security headers to prevent arbitrary code execution [49].

Overall, implementing these techniques mitigates against UFU attacks and ensure that only safe and authorized files are allowed to be uploaded to the system. However, it is important to note that there is no silver bullet solution for secure file upload, and web applications should continually monitor and update their file validation processes to stay ahead of new attack vectors. File size limiting/validation can also be added to the list to prevent a DoS attack on the server, but file size-based attacks are out of the scope of this paper. Finally, we note that there is at least one open source Node.js package for each objective, and if a web application satisfies these objectives, it would prevent all of the thirteen attacks explained in Section 3.2.

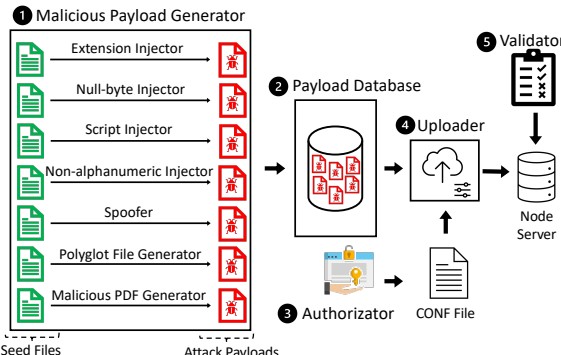

**Figure 2: The architecture of NODESEC and experiment setup to test the security of file upload vulnerabilities in Node.js applications.**

## 3.4 NODESEC

To systematically analyze file upload security weaknesses in Node.js applications and libraries, we developed NODESEC, which automatically generates attack payload files for our attacks (A1-A13) and uploads the malicious payloads to the target web application in an automated fashion. The architecture of NODESEC is depicted in Figure 2 and consists of five main modules. The first module, ❶ Malicious Payload Generator, creates payload files to trigger the attacks. It comprises multiple attack generator components that transform seed files into attack payloads. The second module, ❷ Payload Database, serves as a comprehensive repository for attack payloads. ❸ For authorization purposes, the Authorization module automates the collection of essential data, such as login credentials, cookie tokens, and request headers. The ❹ Uploader module then automates the process of uploading attack payload files to the server side of the target application. Lastly, the ❺ Validator module conducts the validation process after each payload is successfully uploaded. Due to space limitations, we give the full implementation details of NODESEC in Appendix Section 9 and usage of NODESEC in our demo website [50]

## 4 ANALYZING LIBRARIES & APPLICATIONS

In this section, we analyze the security of file upload libraries and real-world web applications written in Node.js using NODESEC. To create our experimental setup, we first downloaded packages from npm containing file upload libraries and real-world web applications, then created sample web applications on our local server. Next, we executed NODESEC to evaluate the security of these real-world web applications and file upload libraries.

For server-side file upload libraries, we adhered to the guidelines provided in their official documentation and employed the widely-used Express.js framework on the server side incorporating libraries as middleware [51]. To assess the libraries' security against UFU, we enforced all available security settings and conducted checks for potential UFU attacks and possible implementation mistakes. We further discuss the functionality of these settings and their effectiveness against UFU. Following a similar approach, we installed

real-world web applications according to the instructions provided in their official documentation. Subsequently, we assessed the security of these sample real-world web applications without modifying their source code.

## 4.1 (In)Securities in File Upload Libraries

We selected the popular server-side file upload libraries based on the following criteria: (1) It should be implemented in Node.js. (2) It should have more than 1K stars on GitHub or received over 80K weekly downloads from npm. Based on this criteria, we selected the most popular six server-side file upload libraries, which, on average, have 2 million weekly downloads from the npm. We analyzed these libraries with NodeSec to see whether they satisfy the file upload validation objectives presented in Section 3.3. We examined both the availability and effectiveness of the security checks. We summarize the results in Table 1.

**Poorly Handling File Name.** To fulfill the file upload validation objective, a library must handle the file name safely. A library can achieve this either by sanitizing or randomizing the name of the uploaded file. While libraries such as multer, formidable, connect-multiparty, and skipper adopt file name randomization techniques, express-fileupload and graphql-upload employ file name sanitization methods to mitigate file name-based attacks. Our analysis revealed that some popular file upload libraries such as express-fileupload and formidable were improperly implementing their functions related to the uploaded file name. In express-fileupload, implementation mistakes in its options cause both the upload of hidden files to a web application server and incorrect trimming of file extensions. In formidable, due to a regex implementation issue, the function fails to correctly parse extensions with multiple dots and does not sanitize characters between the dots, leaving the malicious payload exposed on the application server. Please the demo videos of these issues [3] [4]. Due to space limitations, we present these issues in detail in Section 10 in Appendix.

**Insufficient File Type Validation.** As we defined in our second objective, a secure file upload implementation must validate the expected file type correctly to prevent UFU attacks. Our analysis revealed that all of the file upload libraries are performing MIME type validation for file type validation. While this technique can prevent the executable file upload attack, it can be evaded by performing spoofing attacks [19]. Hence, web applications using one of these popular file upload libraries without any additional prevention method would be insecure to all file type-based attacks.

**No Malicious Content Sanitization.** An adversary can insert malicious content into a seemingly benign file to trigger the UFU vulnerability in the web application. To prevent that, web applications can either sanitize the malicious content in the file or detect and prevent the upload of the malicious file. Nevertheless, our analysis revealed that none of the file upload libraries in our dataset provides a mechanism to detect or sanitize any malicious content-embedded file. Hence, we found that any web application using one of these popular file upload libraries as it is would be insecure to all file content-based attacks.

Table 1: The analysis of popular file upload libraries.

| Library | Version | Weekly Downloads | File Name Validation | File Type Validation | File Content Sanitization |
|---|---|---|---|---|---|
| express-fileupload [52] | 1.2.1 | 276,281 | ◑ | ◑ | ○ |
| multer [53] | 1.4.4 | 4,200,142 | ● | ◑ | ○ |
| formidable [14] | 2.0.1 | 8,736.018 | ◑ | ◑ | ○ |
| connect-multiparty [54] | 2.2.0 | 81,248 | ● | ◑ | ○ |
| skipper [55] | 0.9.1 | 25,672 | ● | ◑ | ○ |
| graphql-upload [56] | 13.0.0 | 358,743 | ◑ | ◑ | ○ |

●: Fully implemented, ◑: Partially/Improperly implemented
○: Not implemented,

## 4.2 (In)securities in Real world Applications

In this section, we present our analysis of real-world Node.js web applications. We utilize NodeSec to perform experiments. We depicted our experiment results on in Table 2. We use output of the NodeSec too fill our table. As explained in Section 2, Node.js servers do not execute files in response to a upload request. So, in our analysis, we did not consider the execution of the file in the server response regarding file execution. As our scope is only analyzing Node.js applications, we consider whether the uploaded file poses a security threat for Node.js application. ✓ indicates that the real-world web application is secure against the attack. In other words, the real-world web application has prevention mechanism against the attack. On the other hand, ✗ indicates that the real-world web application is insecure against the attack. Particularly, the real-world web application has no security mechanism against the attack.

Our investigation throughout this study revealed that mitigating one attack does not necessarily guarantee protection against other attacks within the same category. For instance, during the responsible disclosure process, in one case, after we report the insecurity on real-world web application against Executable File Upload Attack. The maintainer fixed the issue by implementing a MIME-type checking which caused the application to be insecure to other types of file type attacks. This implies that each attack in our attack dataset require a unique consideration, and maybe a specific prevention technique depending on the implementation. Considering these, we decided that grouping the insecurities based on their root causes could potentially underestimate risks and fail to capture variations in the actual exploitation of attacks. Instead, we adopted an approach that treats each successful attack as an individual insecurity. In addition to analyzing real-world web applications with NodeSec, we manually examined their source code to highlight the reasons of these insecurities and good practices currently implemented in these applications. Below, we discuss our findings for each real-world web application. We also provide three demo videos for demonstrate our experiment procedure and usage of NodeSec. [5] [6] [7] [8]

• *GhostCMS:* currently receives over 10k weekly downloads and is used by more than 50k live websites [67]. It uses the multer file upload library to handle the file upload process and it does not use any additional package to prevent UFU. Thus, similar to multer, it

---

[3]Express-fileupload: https://youtu.be/BEcZbZbkjZs
[4]Formidable: https://youtu.be/in1uYJ8tv7M

[5]Tiddlywiki: https://youtu.be/YP7pwBxdpXY
[6]Ghost:https://youtu.be/zzUcpQ2TrWI
[7]Strapi-1:https://youtu.be/JMlxw230ny0
[8]Strapi-2:https://youtu.be/XRRdylZmvtw

| Web Application | Version | Library | File Name-based Attacks | | | | | File Type-based Attacks | | | | | File Content-based Attacks | | |
|---|---|---|---|---|---|---|---|---|---|---|---|---|---|---|---|
| | | | A1 | A2 | A3 | A4 | A5 | A6 | A7 | A8 | A9 | A10 | A11 | A12 | A13 |
| Strapi [57] | 4.1.7 | formidable | ✗ | ✓ | ✓ | ✓ | ✓ | ✗ | ✗ | ✗ | ✗ | ✗ | ✗ | ✗ | ✓ |
| GhostCMS [20] | 4.42.0 | multer | ✗ | ✓ | ✓ | ✓ | ✓ | ✗ | ✗ | ✗ | ✗ | ✓ | ✗ | ✗ | ✗ |
| payloadCMS [58] | 0.15.1 | express-fileupload | ✗ | ✓ | ✓ | ✓ | ✓ | ✗ | ✗ | ✗ | ✗ | ✗ | ✗ | ✗ | ✗ |
| ButterCMS [59] | 1.2.9 | formidable | ✗ | ✓ | ✓ | ✓ | ✓ | ✗ | ✗ | ✗ | ✗ | ✗ | ✗ | ✗ | ✗ |
| Keystone [60] | 4.2.1 | graphql-upload | ✓ | ✓ | ✓ | ✓ | ✓ | ✓ | ✓ | ✓ | ✓ | ✓ | ✗ | ✗ | ✓ |
| Apostrophe [61] | 3.17.0 | connect-multiparty | ✓ | ✓ | ✓ | ✓ | ✓ | ✓ | ✓ | ✓ | ✓ | ✓ | ✓ | ✓ | ✓ |
| Wikijs [62] | 2.5.2 | multer | ✓ | ✓ | ✓ | ✓ | ✓ | ✓ | ✓ | ✓ | ✓ | ✓ | ✓ | ✓ | ✓ |
| Sanity [63] | 2.29.3 | Custom | ✓ | ✓ | ✓ | ✓ | ✓ | ✓ | ✓ | ✓ | ✓ | ✓ | ✓ | ✓ | ✓ |
| FireCMS [64] | 1.0.0 | Custom | ✗ | ✓ | ✓ | ✓ | ✓ | ✗ | ✗ | ✗ | ✗ | ✗ | ✗ | ✗ | ✗ |
| Tiddlywiki [65] | 5.2.2 | Custom | ✗ | ✓ | ✓ | ✓ | ✓ | ✗ | ✗ | ✗ | ✗ | ✓ | ✗ | ✗ | ✗ |
| totaljs [66] | 4.0.0 | Custom | ✓ | ✓ | ✓ | ✓ | ✓ | ✓ | ✓ | ✗ | ✓ | ✗ | ✗ | ✓ | ✗ |

✓: Secure, ✗: Insecure

**A1**: File Extension Injection; **A2**: Null Byte Injection; **A3**: Script-named file name; **A4**: Path Traversal via inserting Non-Alpha; **A5**: Overwrite Attack; **A6**: MIME Type Spoofing; **A7**: Modified Magic Header Bytes; **A8**: JS+JPEG Polyglot **A9**: HTML+PDF Polyglot; **A10**: Executable File Upload Attack; **A11**: JavaScript Embedded PDF; **A12**: PDF Bomb Attack; **A13**: SVG File Upload Attack.

is also insecure to four file type-based and content-based attacks. Interestingly, although Multer randomizes the file name by default, instead of using Multer's randomization function, GhostCMS implements its own custom sanitization function, which causes it to be insecure against A1 attack.

> **✗ Not using already existing function in the library:**
> Our analysis reveals that despite the availability of a security function in the file upload library, an ineffective implementation of a custom validation function can introduce security flaws.

• *Tiddlywiki:* is another popular open-source interactive wiki-like website builder. Although Tiddlywiki implements a file name filtering function to prevent four types of file name-based attacks, it relies on the MIME type for validating the type of the file which resulting in Tiddlywiki being insecure against all types of file type-based and content-based attacks.

• *PayloadCMS:* released in 2021 and received over 12k stars from GitHub [58] and receives over 14k weekly downloads. It makes use of the express-fileupload library to process files uploaded to the server. Our examination of its source code revealed that the security options available in express-fileupload has not been utilized at all. In addition, PayloadCMS has a custom `getSafeFileName` function, which causes it to be insecure to A1 attack. However, it is also insecure to SVG upload attack and Executable File Upload since it does not utilize any sort of file validation method.

> **✗ Missing edge cases in the custom implementation:**
> Our analysis highlights that a custom security implementation must address all edge cases. Consequently, they may defend against certain attacks within an attack category, yet remain susceptible to others from the same group.

• *Strapi:* is the second most popular headless CMS in the top 1M sites [68]. It utilizes the formidable library to process files uploaded

to the server. Our investigation shows that Strapi uses the formidable library without utilizing any additional security mechanism to prevent all the attacks we consider in this work, except for the SVG upload attack. Thus, same as formidable, Strapi is insecure to all types of file type-based and two types of content-based attacks. Different from formidable, it uses a security package named `koa-helmet` module to prevent the code execution in the browser which makes resilient against SVG Upload attack [49].

> **✗ Blind-trust to the file upload library:**
> Our analysis disclose that some application developers may place undue trust in the security features provided by upload libraries, which can lead to overlooked insecurities.

• *Apostrophe* is a popular website builder [61] with currently 4.3k weekly downloads and having received over 4K stars on GitHub. It uses the file upload library connect-multiparty as a file upload library. We found Apostrophe is resilient to all attacks we tested against. The reason is that, similar to wikijs, Apostrophe uses additional security packages such as `Imagemagick` [45] to validate file type. Thus, it is resilient to all types file type-based attacks. Also, it makes use of `sanitize-html` [47] package to sanitize malicious HTML payload inside files. Therefore, it is resilient against the SVG upload attack.

• *Wikijs:* is a Wikipedia-like informative website builder with over 2.8k weekly downloads and over 22k stars on GitHub, [62]. It uses the multer library to handle the file upload process. We found that it is resilient against all types of UFU attacks. We analyzed its source code and found that developers of wikijs utilize different types of open-source security packages to prevent all types of attacks. Particularly, wiki's uses `sanitize-filename` package [41] for handling uploaded file names, `file-type` package package [69] to determine the file type of the uploaded files. Finally, to prevent the code execution via SVG Upload attack, it utilizes `xss` [48] package.

- *Sanity:* is a popular open-source CMS that receives more than 72k weekly downloads [63]. It uses custom functions to handle the uploaded files before they are transmitted to its backend. To prevent file name-based attacks, it assigns a random file name to any uploaded file, which makes sanity resilient to all types of file name-based attacks. Moreover, it validates file type from the metadata of the uploaded file by utilizing the `exif-js` [44] package and it is resilient against all types of spoofing-based, polyglot file, and PDF file attacks. Similarly to Apostrophe, it uses an additional security package, sanitize-html [47], to prevent the SVG file upload attack.

> ✓ Utilizing custom functions and packages to prevent UFU:
> Our analysis showed that demonstrates that a combination of custom implementations and open-source packages can effectively prevent all attacks, while also providing the flexibility to defend against specific attacks as needed.

In addition to these seven real-world applications, we also analyzed ButterCMS, FireCMS, total.js, and Keystone. Their results are presented in Table 2. Due to the page limit, we present their detailed explanation in Section 12 in Appendix.

## 5 ROOT CAUSES & RECOMMENDATIONS

In this section, based on the knowledge gained from our responsible disclosure process and our experiments we enumerate primary factors contributing to the challenges faced by file upload library and Node.js application developers.

**Security Documentation is Imperative.** Our analysis identified a lack of well-designed and comprehensive security documentation for current file upload libraries. This deficiency is a significant challenge for security-unaware developers, as they may be unaware on these issues. While developers are encouraged to incorporate such ready-to-use third-party software packages into their systems, due to missing security documentation, they are unfortunately left with minimal guidance on how to implement security against UFU attacks. Our analysis of real-world web applications concluded that the absence of security documentation on file upload libraries places application developers to an uncertain position, where they inadvertently introduce security misconfigurations to their applications or fail to implement necessary file validation/sanitization mechanisms which makes their applications insecure against UFU attacks. Consequently, library developers should explicitly enumerate the use cases of the file upload library and clearly state both the absent and present security mechanisms against UFU attacks in their security documentation.

**Lack of Consensus of Responsibility.** Our analysis elucidates that the process to report and patch critical libraries is not entirely clear to the involved parties. As our disclosure process exhibits an absence of a well-defined consensus concerning the allocation of accountability for implementing content and metadata validation within the file upload ecosystem. While some library developers took the reported issues very seriously and initiated the patching process, others conveniently deferred the responsibility to web application developers. This confusion on where the responsibility lies greatly endangers the overall security of the file upload ecosystem. Consequently, we posit that the responsibility to implement defense against UFU should be clearly defined and we suggest that

the implementation of security checks at the file upload library level would be scalable and substantially contribute to mitigating the risks associated with this critical issue in the file upload ecosystem. **Lack of Comprehensive Security Test Cases.** Our analysis disclosed that popular file upload libraries (e.g., express-fileupload, formidable) contain bugs within their security-related functions. As we detail in Section 10, these bugs not only present theoretical risks but also attack vectors that can be exploited by malicious actors to launch various types of UFU attacks on web applications utilizing these libraries. Our analysis on the source code of these libraries revealed that they have not been tested against all the attack scenarios that can be utilized by an attacker. For example, the sanitization function in the formidable library did not consider payloads that can be inserted between the extensions of the uploaded files. Consequently, our recommendation is the developers must consider all the edge cases while implementing these functions in their libraries.

## 6 RELATED WORK

**Node.js Security.** SYNODE [70], a static analysis-based prevention tool against injection attacks in the Node.js ecosystem. In [71], the first security architecture for Node.js, was introduced. In [72] investigated the security of Node.js applications against ReDoS attacks. Differently, Nielsen et al. [73] proposed a modular call graph-based approach for security scanning in Node.js applications. In [74], the authors focused on the communication process between client- and server-side code in Node.js programs and identified vulnerabilities that can lead to different types of server-side attacks. **File Upload Security.** Huang et al. proposed UChecker [7], a static analysis-based tool that automatically detects UFU vulnerabilities in PHP-based server-side web applications. Likewise, Huang et al. introduced UFuzzer [8], a locality analysis-based UFU vulnerability detection system for PHP applications. In [6], the authors proposed a penetration testing tool for identifying file upload bugs in PHP-based web applications. While

**Differences from existing work.** While there exists a substantial amount of prior work investigating the security of Node.js, none focused specifically on the UFU attacks. Additionally, although there are studies that propose tools to detect UFU vulnerabilities in PHP web applications, there is no prior work that investigates both the security of file upload libraries and the applications in Node.js ecosystem. We also enumerate the challenges in using existing tools for Node.js and compare the attack coverage of NODESEC with existing tool in Section 13 in Appendix.

## 7 CONCLUSION

In this paper, we analyzed the security posture of the file upload ecosystem in Node.js. We introduce NODESEC a tool designed to analyze Node.js applications and libraries in the context of UFU-related security issues and we analyze popular file upload libraries and real-world web applications using NODESEC. Our analysis revealed security issues in eleven popular Node.js web applications and bugs in file upload libraries and received 19 CVEs. With this study, we aim to raise awareness about the importance of security measures in file upload libraries and web applications, contributing to the development of better practices and tools to protect users from UFU attacks.

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

# APPENDIX

# 8 VULNERABILITY DISCLOSURE AND RESPONSES

We responsibly disclosed our findings to the respective developers and maintainers of the libraries and CMSs. As of writing this paper, we received 8 CVEs for libraries and 11 CVEs for CMSs. For disclosure, we sent an initial notification email to each developer. We sent a second email to the ones that responded to our notification email. In the second email, we included the following: (i) general description, (ii) implementation issues, (iii) steps for reproducing the vulnerabilities, (iv) the proof-of-concept attack videos, and (v) potential countermeasures for each vulnerability. The summary of library and CMS developers' responses is given in the Table 3.

**Table 3: Library/CMS developer responses and patching status of the reported vulnerabilities.**

| Name | Type | Version | Response | Ack | CVE Issued | Patching Status |
|------|------|---------|----------|-----|------------|-----------------|
| express-fileupload | Library | 1.2.1 | Yes | Yes | Yes | Yes |
| multer | Library | 1.4.4 | Yes | No | Yes | No |
| formidable | Library | 2.0.1 | Yes | Yes | Yes | Yes |
| connect-multiparty | Library | 2.0.0 | No | No | Yes | No |
| skipper | Library | 0.9.1 | Yes | No | Yes | No |
| graphql-upload | Library | 13.0.0 | Yes | No | Yes | No |
| GhostCMS | CMS | 4.42.0 | Yes | Yes | Yes | No |
| payloadCMS | CMS | 0.15.1 | Yes | Yes | Yes | No |
| Strapi | CMS | 4.1.7 | Yes | Yes | Yes | Yes |
| ButterCMS | CMS | 1.2.9 | Yes | Yes | Yes | Yes |
| Keystone | CMS | 4.2.1 | No | No | Yes | No |
| FireCMS | CMS | 1.0.0 | No | Yes | Yes | Yes |
| Tiddlywiki | CMS | 5.2.2 | Yes | Yes | Yes | No |
| totaljs | CMS | 4.0.0 | No | No | Yes | No |

**Library Developers' Responses and Reactions.** We received an initial response from five libraries within three days after the initial notification email. We did not receive a response from the connect-multiparty library. Meanwhile, express-fileupload and formidable libraries acknowledged the issues and implementation mistakes related to their file name handling on uploaded files. They fixed the problems and released patched versions. Moreover, although the multer library acknowledged the issue, they refused to patch the library, claiming that their library simply accepts all files regardless of their content and deferred the responsibility to web application developers to apply the necessary security checks. Furthermore, the graphql-upload library considered the reported issue

to be outside the scope of the offered service since the library operates as middleware. We recommended that they clarify this in their security documentation, stating that they do not perform any security checks in their library to avoid misleading web application developers.

**CMS Developers' Responses and Reactions.** As for the responses from the real-world web applications, four out of the eight vulnerable real-world web applications analyzed, namely Strapi, ButterCMS, GhostCMS, and PayloadCMS, replied our notification email in two weeks. So far, ButterCMS and Strapi fixed the vulnerabilities. On the other hand, GhostCMS and PayloadCMS refused to patch the vulnerabilities. They claimed that all users were considered to be trusted in their "threat model", and that they did not expect them to upload malicious files. Furthermore, totaljs [66] requested more information about our findings and we sent the aforementioned detailed attack description email. However, we have not received any reply as of the writing of this paper. At the time of this writing, we unfortunately still have not heard back from the developers of fireCMS, tiddlywiki, and keystone for the vulnerabilities notified.

**US-Cert Involvement.** We also notified the US-CERT about the issues. They acknowledged our findings and decided to create two cases, one for the libraries and one for impacted CMSs.

# 9 IMPLEMENTATION DETAILS OF NODESEC

To systematically analyze the file upload security weaknesses in Node.js applications and libraries, we developed NodeSec, which automatically generates attack payload files for our attacks (A1-A13) and uploads the malicious payloads to the target web application in an automated fashion. The architecture of NodeSec is depicted in Figure 2 and consists of the following main modules: 1) Malicious Payload Generator, 2) Payload Database, 3) Authorization 4) Uploader, and 5) Validator.

**Malicious Payload Generator.** This module creates payload files to trigger the attacks detailed in Section 3.2. It comprises multiple attack generator components that transform seed files into attack payloads. After generating the payloads, we verify whether they are executable following the modification and ensure the preservation of their semantics. Its components include:

- *Malicious Extension Injector.* This component takes a seed file as input, such as test.js, and injects multiple extensions through either random prepending or appending with benign extensions (e.g., test.js.png, test.png.js), removing the file extension entirely (e.g., test), disguising the file extension by randomly altering the case of its characters (e.g., test.Js, testJS), and appending unusual extensions (e.g., seed.html5, test.js6). It also generates filenames with triple-appended extensions (e.g., seed.pdf.html.png), randomly mixed case extensions (e.g., test.Js, test.hTml.jPEg), and unusual extensions (e.g., jsx, mjs, xhtml).

- *Null-byte Injector.* This module inserts null-byte characters at random positions within the file name. To enhance the attack coverage, we base this component on the set of files generated by the Malicious Extension Injector. Specifically, this component places different null-bytes in random positions in the file name. For example, this module creates file names such as test.js%00.png, test.js%.png.

- *Script Injector.* This component receives a seed input file and injects script payloads at random positions within the file name. Similar to the Null-byte Injector component, this module increases the attack coverage of our tool by injecting payload scripts into random positions of the file name. For instance, using this module, a JavaScript seed payload file named "test.png.js" can be transformed into "test.png[payload].js".
- *Non-alphanumeric Injector.* This component takes a seed input file and injects malicious non-alphanumeric characters into random positions in the file name. For example, given a valid PNG file named "file.png" as input, the component can generate file names such as "/../..png".
- *Spoofer.* This component takes an input file and generates a spoofed version with either altered MIME type or magic bytes. It consists of two separate functionalities: MIME type spoofing and magic-byte spoofing. For MIME type spoofing, the component reads a JSON file containing a list of MIME types and their corresponding file extensions. It iterates through the input files and alters their MIME type from the original value (e.g., "text/javascript") to a different MIME type (e.g., "application/pdf") while keeping the file content unchanged. The output files are saved with the same file extension but with an updated MIME type, potentially bypassing file type validation checks based on MIME types. In the case of magic-byte spoofing, the component reads a JSON file containing a list of magic bytes associated with different file formats. It iterates through the input files and modifies the magic byte of each file with another file format. The output files are saved with a new file extension corresponding to the spoofed magic byte, while the file content remains the same.
- *Polyglot File Generator.* This component creates polyglot files, specifically PDF+HTML and PNG+JS combinations. Polyglot files are files that are valid in multiple file formats, allowing them to bypass certain file validation mechanisms and introduce security risks in web applications [32]. To generate PDF+HTML polyglot files, the script creates a simple PDF file using the ReportLab library and a basic HTML file with a heading. It then combines the PDF and HTML data, separated by a custom delimiter, into a single file. For PNG+JS polyglot files, the script reads an existing PNG file, calculates its header size, and injects a JavaScript payload, preceded by a sequence of null bytes, into the file without affecting its validity as a PNG image.
- *Malicious File Generator.* This component generates various types of malicious files by embedding payloads within benign PDF, SVG, and HTML files. The generator creates JavaScript-embedded and compressed PDF files, SVG files with different payloads, and modified HTML and EML files.

**Payload Database.** This component serves as a comprehensive repository for attack payloads, which are then uploaded to the target web application through the Uploader module. In addition to the payloads generated by the Malicious Payload Generator for our attacks considered in this study, the users can employ their own scripts to generate different payload files or directly import pre-created payload files to the database.

**Authorization Module.** The Authorization module is a Node.js script that automates the collection of essential data, such as login credentials, cookie tokens, and request headers. The script prompts the user for required details such as login URL, upload target URL, upload directory, username, and password. After, it navigates and extracts the session cookie and headers, creates a configuration object, and saves it as a JSON file. The configuration file is employed by the Uploader module to automate the upload of payload files.

**Uploader.** The Uploader module automates the process of uploading attack payload files to the server side of the target web application. This module necessitates two critical input files: 1) a configuration file retrieved from the Authorization module and 2) a payload file designated for uploading to the target web application. It employs the `request-promise` [75] library to generate upload requests and accepts configuration and payload files as inputs. The module includes necessary functions that automate the upload process for different payload files such as preparing form data.

**Validator.** The Validator module conducts the validation process after the successful upload of the payload for each attack. For attacks involving malicious characters or extensions in the file name (A1-A5), this component examines the web application's sanitization process. It determines whether the file name has been adequately sanitized by checking the name of the uploaded file. If the file name retains any malicious characters and/or patterns after a successful upload, the Validator module classifies the web application as vulnerable. To validate the attacks that could result in code execution on Node.js servers or browsers (A6-A13), the module checks the file contents by searching for malicious content signatures. For example, to validate the attack A13, the module scans the SVG file and checks for a malicious script. After validating the presence of malicious content, it remotely executes the file using the Node.js interpreter v16.14.0 by employing Node.js script. The execution is performed by a script that accesses the uploaded payload file's path via the URL obtained from the target web application's Node server.

## 10 CASE STUDIES

In this subsection, we present two case studies that demonstrate the efficacy of NODESEC in detecting implementation errors in file upload libraries. Our all findings were acknowledged and patched by the developers of the libraries.

**Express-fileupload.** Express-fileupload is a popular file upload library for Node.js applications. It receives almost 284,620 weekly downloads and is used by 169k open-source projects [52]. To prevent file name-based attacks, the library contains security options such as `SafeFileNames` and `preserveExtension` to sanitize the name of the uploaded file to make the web application resilient against file name-based attacks. While analyzing the security of this library with NODESEC we found that these functions were incorrectly implemented. During our experiments, we saw that while uploading the payload file named `/../../.html`, which contained only non-alphanumeric characters, the library uploaded the file as `.html`. The manual analysis of the source code revealed that the `SafeFileNames` does not properly handle the non-alphanumeric characters in the file name. As demonstrated in Lines 14-25 at Listing 2, it strips all the non-alpha characters in the file and constructs the filename by concatenating the name and extension. This construction preserves that dot in the file name and thus allows

hidden files to be uploaded to a web application server. Such behavior introduces security weaknesses to the library as filenames beginning with a dot are considered hidden in UNIX-like systems. This weakness can be exploited by adversaries to upload malicious payload files to a web application server. Furthermore, while the preserveExtension option is designed to set the extension's length, our experiments using NODESEC disclosed a potential abuse by attackers. During the experiment, NODESEC generated a malicious payload named test.4mjs. After the upload, the payload's filename was renamed to test4.mjs. As demonstrated in Listing 2 due to an implementation issue, the extension is not trimmed correctly in cases where the extension length is bigger than the settled max length. As demonstrated in our demo website. This implementation mistake could be leveraged by attackers to upload malicious payloads onto a web application server and trigger shell code execution.

**Formidable.** Formidable is a popular file upload library that receives nearly 9 million weekly downloads and is used by 1.5 million open-source packages [14]. While testing the formidable file upload library against file name-based attacks with NODESEC, we observed that the library's file name sanitization function was improperly implemented. The manual analysis of the code revealed that the getExtension function does not handle multiple dots in file names correctly. Due to the regex implementation issue as demonstrated in the Listing 3, the function fails to correctly parse extensions with multiple dots (e.g., .png.html) and does not sanitize characters between the dots. For example, when uploading a payload such as test.png[payload].html, the library sanitizes the file name until the first extension, leaving the malicious payload exposed on the application server. This could potentially allow an attacker to bypass security measures implemented by the web application.

## 11 ATTACK EXECUTION AND CONSEQUENCES

Evading security checks in a web application to upload a malicious payload is itself unintended and exposes the application's insecurity. However, to successfully exploit the vulnerability, as explained in our threat model in Section, the attacker still needs to find a way to execute the uploaded payload, either on the client or server side. We discuss these methods and their potential consequences below.

**Executing on the client side.** The file name or content reflected by the web page can cause arbitrary code execution on the client side. The attacker can employ various tactics to trigger code execution on the client side, such as directly uploading an HTML or JS file to a web page. Then, a user can trigger the execution of the file by accessing or opening it from the public path of the file [76]. This can lead to various attacks, such as stealing sensitive user data or redirecting users to malicious websites.

**Executing on the server side.** Unlike PHP, Node.js compiles JavaScript code into machine code before execution, to minimize the arbitrary code execution [77]. Nevertheless, the attackers can still exploit UFU vulnerabilities to execute code in the server-side. For instance, attackers may exploit dangerous Node.js functions, like eval() and exec(), to enable server-side execution of uploaded files. For example, after the malicious payload file is uploaded to the server, the eval() function implemented on the server side

```
1  const parseFileNameExtension = (preserveExtension,
       fileName) => {
2    // ...
3    const nameParts = fileName.split('.');
4    if (nameParts.length < 2) return result;
5
6    let extension = nameParts.pop();
7    // ISSUE: The extension is not trimmed correctly when
         its length is equal to maxExtLength
8    if (extension.length > maxExtLength && maxExtLength >
         0) {
9      // ...}
10   result.extension = maxExtLength ? extension : '';
11   result.name = nameParts.join('.');
12   return result;
13 };
14 const parseFileName = (opts, fileName) => {
15   // ...
16   parsedName = uriDecodeFileName(opts, parsedName);
17   if (!opts.safeFileNames) return parsedName;
18
19   const nameRegex = typeof opts.safeFileNames === 'object
         ' && opts.safeFileNames instanceof RegExp
20     ? opts.safeFileNames
21     : SAFE_FILE_NAME_REGEX;
22   let {name, extension} = parseFileNameExtension(opts.
         preserveExtension, parsedName);
23   if (extension.length) extension = '.' + extension.
         replace(nameRegex, '');
24   // ISSUE: The following line allows uploading hidden
         files (starting with a dot)
25   return name.replace(nameRegex, '').concat(extension);};
```

**Listing 2: A sample code snippet from the express-fileupload library.**

```
1  _getExtension(str) {
2    const basename = path.basename(str);
3    const firstDot = basename.indexOf('.');
4    const lastDot = basename.lastIndexOf('.');
5    //ISSUE:Doesn't handle multiple dots in extension.
6    const extname = path.extname(basename).replace(/(.[a-
         z0-9]+).*/i, '$1');
7    if (firstDot === lastDot) {return extname;}
8    // ISSUE: Doesn't sanitize characters between dots
9    return basename.slice(firstDot, lastDot) + extname;}
```

**Listing 3: A sample code snippet from the formidable library demonstrating the improper handling of multiple dots and lack of sanitization in its function.**

can execute a JavaScript code embedded in the payload file. While the security sandboxing of JavaScript decreases the dangers/risks of these functions by preventing the execution of the code in the browser, Node.js does not have a built-in security sandbox [17, 70]. In some cases, the reliance on third-party libraries and modules in Node.js applications can introduce vulnerabilities. For example, after uploading a malicious payload file, an attacker could exploit an insecure implementation of the security-related function (e.g., sanitization function) in a third-party library, executing the embedded JavaScript code in the payload file [15].

# 12 ANALYSIS OF MORE REAL-WORLD WEB APPS

• *ButterCMS:* is another popular CMS with over 20k weekly downloads [59]. It employs the formidable library for processing the uploaded files. Our analysis, conducted with NODESEC, reveals that it does not use any additional security mechanisms to strengthen its file upload security. As a result, ButterCMS is vulnerable to the same set of attacks as the formidable library, which includes file type-based (A6-A10) attacks and content-based attacks.

• *FireCMS:* is a CMS used by various websites from different sectors and received nearly 1k stars on GitHub [64]. It utilizes custom-implemented functions to process the uploaded files before sending them to the server. Thanks to its custom build `fileNamebuilder` function, FireCMS is resilient to four types of file name-based attacks. Nevertheless, it does not utilize any type of file validation mechanism to prevent the uploading of malicious files which results in FireCMS being insecure to other types of attacks.

• *Totaljs:* is an open-source CMS with more than 1.7M downloads [66]. It uses custom file upload functions to process the uploaded files. We found that it is resilient against all types of file name-based attacks since it assigns a random name to an uploaded file via a custom-implemented function. Nonetheless, it does not utilize any type of file validation technique before uploading a file to its server. Thus, it is insecure against the A10 attack type. However, it performs pre-processing and resizing operations on the images before displaying them on the front end. In this process, it raises an exception while pre-processing the image files with a payload. Thus, it is resilient against three types of file type-based attacks and one type of content-based attack. However, it is not resilient against another type of file type-based and two types of content-based attacks.

• *Keystone:* is a popular CMS with over 1.7k weekly downloads [60]. It makes use of graphql-upload for uploading files to the server. Our analysis showed that Keystone implements additional security mechanisms by assigning a safe file name to an uploaded file before sending it to the server by using `filenamify` [42] package, which makes it resilient against all types of file name-based attacks. Moreover, it uses the `image-type` module [78] to determine and validate the file types of images, which prevents three types of file type-based attacks and SVG upload attacks. Nevertheless, it does not utilize any mechanism to detect malicious content within PDF files. Hence, Keystone is only insecure against two types of content-based attacks.

# 13 COMPARISON WITH EXISTING TOOLS

The fundamental concept of file uploading remains consistent (i.e., transferring a file from a client to a server) among different server-side technologies. So, theoretically, all the existing tools can be adapted for analyzing Node.js applications. Whereas, this requires significant domain expertise and in-depth knowledge of the existing tools' source. Below, we enumerate the challenges in using existing tools for Node.js and compare the attack coverage of NODESEC with existing tools.

# A Challenges in Using Existing Tools for Node.js

During our experiment, we observed the main challenges of using existing tools for Node.js applications as follows:

**Syntax and Structure.** The syntax, structure, and functions of programming languages can differ significantly, leading to challenges in detecting and addressing UFU vulnerabilities across different languages. For example, PHP uses a mixture of tag-based and imperative syntax, while Node.js adopts a more uniform object-oriented and functional approach [22]. This difference in syntax and structure makes static analysis techniques tailored for PHP syntax, such as those employed by UChecker [7] and UFuzzer [8], inapplicable for Node.js applications. For instance, the lack of tag-based syntax in Node.js might influence the detection process, as tools designed for PHP do not parse or analyze JavaScript code to detect UFU vulnerabilities. Moreover, the execution of listeners within third-party packages in Node.js applications is event-driven, which can pose challenges for static analysis-based approaches [79].

**Library Usage and Different File Handling.** PHP-based web applications commonly use the `_FILES` superglobal array and built-in functions, such as `getimagesize()` and `finfo_file`, during the file upload process [22]. These built-in functions are part of the core PHP language. So, they inherently provide a safer setup. Conversely, Node.js web applications frequently use third-party file upload libraries. Hence, when analyzing UFU vulnerabilities in Node.js applications, it's vital to create attack payloads that specifically address edge cases in these libraries. We evaluated the existing tools on the formidable library and found that the techniques employed by these tools were unable to detect the existing implementation flaws since their attacks mostly focus on the lack of checks during the file upload mechanisms of the web application. On the other hand, as detailed in Section 4.1, NODESEC uniquely generates file names by inserting payloads at random positions, which enabled it to exclusively identify the formidable library's implementation error.

**Different Execution Environments.** PHP-based web applications typically run on web servers like Apache or Nginx, whereas Node.js-based web applications either have their own integrated web server or use a server like Express.js [51]. Our experiments have shown that existing tools, such as FUSE [6] and Fuxploider [80], generate payloads based on PHP tags and PHP-specific functions in their payload generation process. These tools are designed to detect UFU vulnerabilities in PHP web applications, and their payloads are intended for execution within PHP interpreters on Apache or Nginx servers or specific server configurations. Although these payloads can successfully identify distinct UFU vulnerabilities in PHP applications, they are not applicable to Node.js applications. We observed that FUSE-generated payloads such as 'seed.PHP,' incorrectly labeled the application as vulnerable. However, Node.js applications are not designed to process PHP code, which results in false positives. Furthermore, existing file upload analysis tools primarily focus on the execution of JavaScript files on the client side. In contrast, Node.js environments also allow for the execution of JavaScript files server-side [70].

## B  Attack Coverage

In this section, we compare NODESEC's attack coverage with other tools.

**FUSE.** is designed to identify UFU vulnerabilities in PHP-based web applications by performing 13 different mutation techniques (M1 to M13) on seed files such as HTML, JS, XHTML, and PHP. NODESEC covers mutations M1, M2, M3, M6, M8, M9, and M13. The mutation M5 replaces PHP tags in PHP files, is not applicable to Node.js applications that do not use PHP tags. Mutations M4, M7, M10, M11, and M12 in FUSE are designed to modify the name of an uploaded file, all of which are covered by NODESEC. Moreover, unlike FUSE, NODESEC considers several other attacks related to file name and file type, such as null-byte injected file name, script-named file name, path traversal, and PDF bomb attack.

**UploadScanner.** allows testing of web application security mechanisms against both file type-based and file content-based attacks. To evaluate web applications against file name-based attacks, Upload-Scanner generates payloads by prepending an extension to a file name, changing the file's extension, and injecting scripts, null-bytes, and path traversal payloads into the file name. NODESEC enhances these attacks by integrating randomization logic and inserting null bytes, scripts, and path traversal payloads into random positions of the file names. This enables the generation of more sophisticated payloads, such as test.js[payload].png. Notably, with this approach, NODESEC uncovered 3 different implementation mistakes in the popular file upload libraries.

**Fuxploider.** is an open-source tool that automates the detection and exploitation of file upload vulnerabilities. It focuses on detecting UFU vulnerabilities in PHP and JSP applications by employing the issues in specific functions such as the `phpinfo()` function, and specific server configurations like Apache 2.4. In terms of attack coverage, Fuxploider can upload a file by changing its extension to upper and lower case, using an uncommon extension, and altering the MIME-type, all of which are already covered by NODESEC.

In addition to the most related tools mentioned above, there are alternative tools for detecting vulnerabilities in Node.js applications. However, we did not include them in our analysis due to space limitations. For example, OWASP ZAP's FileUpload Add-on [81] is heavily inspired by UploadScanner, while others (e.g., client-side scanners) did not exclusively focus on UFU vulnerabilities [82–84].

*B.1  Configurability and Deployability.* Besides attack coverage, NODESEC advances existing tools in terms of configurability and deployability. First, all of the compared tools [6, 80, 85] are implemented in Python 2, which has reached its end-of-life [86]. Additionally, these tools haven't been actively maintained despite the constant evolution of web technologies, including server-side development platforms. Finally, UploadScanner was developed as an extension for the premium version of Burp Suite Pro, restricting its deployability. Conversely, NODESEC is developed with Python 3 and Node.js, fully open source (all of the modules in Section 3.4), and provides easy configurability due to its modular structure.

