# OpenReview forum: "(In)Security of File Uploads in Node.js"
_ACM.org/TheWebConf/2024/Conference — TheWebConf24_

### Official Review · Reviewer_BfP6 · 2023-10-24

**Novelty:** 7
**Technical Quality:** 6

**Review:**

This paper proposes NODESEC to analyze file upload libraries and real-world applications in the Node.js ecosystem.
The evaluation shows that Unrestricted File Upload is the most impactable vulnerability category in the Node.js ecosystem.
This paper leverages NodeSec to analyze the six most popular server-side file upload libraries, which, on average, have two million weekly downloads.
In the evaluation, this paper selects 13 different attacks in 11 popular real-world web applications, grouped into three categories: (1) File Name-based Attacks (File Extension Injection, Null Byte Injection, Script-Named File Name, Path Traversal, Overwrite), (2) File Type-based Attacks (Spoofing-based, Polyglot File Attacks, Executable File Attacks), and (3) File Content-based Attacks (PDF File Attacks, SVG Upload Attacks).


The evaluation received 19 CVEs and two US-CERT cases.  The CVEs are mainly caused by 1) they overlay on file upload libraries for security, 2) they make errors when configuring security options in the libraries, and 3) they fail to address all edge cases in their custom implementation.

The explanation of the methodology in NODESEC is limited. However, the results and summary are pretty impressive, the takeaways give a good guideline for node js developers.

**Questions:**

The paper addresses 13 attacks in 11 nodejs applications. The attacks are grouped in 3 categories by the attacks. However the paper did not mention what are the potential consequences of each attack. Such as whether these attacks could lead to (1) malicious file uploads: Attackers may upload files containing harmful scripts that could compromise your application or server;  (2) Denial of Service (DoS) attacks: A large number of file uploads can exhaust server resources, causing the app to become unresponsive. OR (3) Sensitive data exposure: Unauthorized users may gain access to files containing sensitive information.

Authors should explain the threat model and conduct risk analysis of each attack, and show how the attacks impacts real world nodejs systems.

**Reviewer Confidence:**

4: The reviewer is certain that the evaluation is correct and very familiar with the relevant literature

**Scope:**

4: The work is relevant to the Web and to the track, and is of broad interest to the community

---

### Official Review · Reviewer_iDGh · 2023-11-03

**Novelty:** 4
**Technical Quality:** 6

**Review:**

**Summary:**
The paper presents a study of Unrestricted File Upload (UFU) security issues in Node.js libraries and popular applications which use the libraries in the study. The authors implement an open source tool NODESEC which generates unique payloads for testing various file upload insecurities. As a part of this effort, the authors receive 19 CVEs for their efforts carrying out responsible disclosure.

**Review:**

As a part of the methodology, the authors identify various well known techniques by analyzing the NVD CVE database relevant to insecure file uploads resulting in 1846 CVEs related to UFU vulnerabilities. The authors systematically group the 13 chosen attacks into three categories, (1) file name based attacks, (2) file type based attacks, and (3) file content based attacks, and provide details of the various attacks in the Attack Descriptions section. Additionally, the authors evaluate popular node.js file upload libraries and applications leveraging these libraries for their vulnerability to the chosen attacks. The paper presents a breakdown into the root causes and ends with recommendations.

The authors argue for the need for using file name and content sanitization techniques either through name validation or identifying metadata in addition to MIME type data to identify the actual type of a file based on the file stream. Given the adversaries ability to trigger an UFU vulnerability resulting in a RCE bug, the authors recommend randomly generated ID assignment to the filename -- a recommendation which is sound for applications such as CMSs leveraging the file upload functionality.

**Originality:**
While the paper is written clearly and the contribution of the NODESEC tool is valuable to the security and the web community, it is unclear that the contributions of the findings in this paper are extremely novel since all the attacks described in the paper are well known due to previous CVEs with similar issues. Similarly while the contribution of NODESEC is valuable, it is an incremental (but important) enhancement over the UploadScanner tools but with integrations for sophisticated payload generation resulting in uncovering 3 different implementation mistakes. Arguably, this gap in the libraries' default capability is understood by a few applications surveyed  such as Apostrophe, Wikijs, Sanity which use insecure libraries but have additional safety checks either implemented through custom means, or by bundling other modules such as a sanitize module for improved security and preventing UFU vulnerabilities.

**Clarity:** The paper is very clearly written with no major typo issues and the tables indicating the results are extremely clear and a delight to read.

**Significance:** As the authors encountered in their efforts during disclosure, there is a lack of consensus of responsibility in who patches their software, while it is important for library developers to present adequate warnings or enumerate the list of possible edge cases which could happen to developers who use their code, the responsibility is equally same for applications which use these libraries. Contributing a comprehensive test case suite which can be generated by the NODESEC tool hopefully makes it much easier for developers to harden their library/applications' security. It would have been interesting to observe correlations between the number of active maintainers for each library/application, and their response to the disclosure, or the extent of their vulnerability shown in Table 2. The Node.js community is well aware of issues such as the leftpad incident [1] which resulted in various failures in Node.js libraries, build systems, across the world. Critical libraries for large applications could be implemented by a small set of "overwhelmed" maintainers and it'd have been interesting to observe any such correlations in this effort which could contribute to stronger recommendations. The current set of recommendations while accurate and important are well known to software developers, stronger insights and more novel recommendations/mechanisms to enforce them would have made this paper much stronger.

Overall, I'd advocate for the acceptance of this paper into the proceedings due to the thoroughness in evaluation, ease of understanding, tool contributions to the community, and clear writing.

[1] https://qz.com/646467/how-one-programmer-broke-the-internet-by-deleting-a-tiny-piece-of-code

**Questions:**

1. Figure 1 on page 2 is slightly misleading because the first arrow between the user and the client side application is an action which is performed on the local interface, the HTTP(s) POST request is not sent as a part of this and is used to communicate between the client to the backend webserver through a POST request sending the form data. Renaming the first arrow between the user and client is more correct. This is a minor change.

2. The authors mention the criteria for selecting server side libraries indicating that the library should have received over 80K weekly downloads from npm, How many libraries were returned matching this constraint? Were the 6 libraries chosen the top 6 libraries or were they the *only* libraries? Additional clarity by mentioning the exact dates when this measurement was conducted is desirable for the completeness and thoroughness of the methodology.

3. The vulnerability disclosure could benefit from additional information such as a timeline of report, response, acknowledgement, CVE issuance and patching for each library shown in Table 3.

**Reviewer Confidence:**

4: The reviewer is certain that the evaluation is correct and very familiar with the relevant literature

**Scope:**

4: The work is relevant to the Web and to the track, and is of broad interest to the community

---

### Official Review · Reviewer_jENr · 2023-11-17

**Novelty:** 5
**Technical Quality:** 4

**Review:**

This is an interesting paper where all sources of UFUs have been identified and this has led to multiple (I think 17) registered CVEs. The work is very solid, comprehensive and robust. The paper is undoubtedly impressive and does a good job in thoroughly characterizing and explaining the problem. Also, the fact that the authors could get CVEs indicates the acceptance in the community.

A couple of things can still be done to improve the paper. They explain why I am slightly lukewarm about the paper.

1. File uploading is still a narrow topic. Perhaps all forms of data uploading should be looked at. Of course, not in the same paper. However, some statements should be made about how the results here translate to a more general setting (POST kind of traffic).
2. What about loading trusted and verified JS libraries? What is the need to use a library when it is known to not be completely secure? Isn't it a better idea to either use a verified library by let's say a major vendor like Google or write a library from scratch? After all, file uploading is not a very complicated thing, as long as the file sizes are relatively small.

**Questions:**

1. Can the authors generalize their results to other forms of data uploading?
2. Have the authors looked at trusted JS libraries?

**Ethics Review Description:**

It is fine by us.

**Reviewer Confidence:**

3: The reviewer is confident but not certain that the evaluation is correct

**Scope:**

3: The work is somewhat relevant to the Web and to the track, and is of narrow interest to a sub-community

---

### Official Review · Reviewer_PLnG · 2023-11-21

**Novelty:** 3
**Technical Quality:** 4

**Review:**

This paper presents a study of the (in)security of the file upload functions in web-based libraries and applications developed atop Node.js. To kick off the study, the authors first crawled the NVD database to understand the threat landscape of file upload. After categorizing the common vulnerability types (which are known as Unrestricted File Upload, or UFU), a measurement pipeline tool is proposed to identify vulnerabilities from existing open-sourced Node.js libraries and applications that involve file upload features. The measurement results revealed a widespread of vulnerabilities, followed by the root causes as well as remedies to mitigate the security issues.

Strengths
- Measurement study on the security uploads among popular Node.js libraries and applications
- An automated measurement pipeline is presented for testing real applications
- Practical UFU vulnerabilities (19 CVEs and two US-CERT cases)

Weaknesses
- Insufficient novelty in the security of file uploads and the identified CVEs
- Vulnerabilities found are generic and not unique to Node.js
- Unclear how comprehensive the study is and how many actual web applications are subject to attacks


Thank you for submitting to TheWebConf’24. I appreciate the fact that the authors have studied practical security problems in web applications, and developed tools to measure and identify security vulnerabilities among them. The analysis yielded fruitful results with 19 CVEs and two US-CERT cases in widely used Node.js applications. However, I still have a number of concerns regarding the paper.

First of all, UFU vulnerabilities have existed for a long history and there have been numerous studies on this issue.  For example, [A] systematically categorizes vulnerabilities in file upload for web applications. Similarly, the paper also conducts a study to understand these vulnerabilities, which appears to be somewhat unnecessary. I suggest the authors cite the referenced paper and remove the duplicated work in crawling the NVD database. Based on this observation, I believe the paper’s contribution is more on the measurement aspect, i.e., to understand how many Node.js libraries and web applications are subject to UFU attacks.

The second major concern is that neither the file upload vulnerabilities nor the proposed methodology are unique to Node.js and bring substantial new insights. In particular, the attack description introduces a number of vulnerabilities, which are generic to the web and other types of applications. The technical design (NodeSec) is mainly about testing the vulnerabilities on a running Node.js application without diving into details about the technical challenges and solutions. In this regard, one may ask the question of whether existing UFU vulnerability identification tools can be directly applied to solve the problem. Thus, the contributions with respect to the technical space and problem space are marginal.

The third concern is that it is unclear how comprehensive the study is. Based on the statistics about the studied subjects, it seems like they are quite popular and have been used in many products. However, it is still unclear how widely they are used from a global view. For example, the paper does not describe the criteria for selecting the 11 Node.js applications in Table 2. Are they selected because they are the top open-source applications?

Minor issues:

- The threat model (i.e., Attack Descriptions in 3.2) could be expanded to include more details about the attacker types, objectives, and assumptions.
- Many abbreviations should be clearly defined when they first appear in the paper, including CMS and MIME.

[A] Uddin, Nasir, and Mohammad Jabr. "File upload security and validation in the context of software as a service cloud model." 2016 6th International Conference on IT Convergence and Security (ICITCS). IEEE, 2016.

**Questions:**

- What are the technical challenges in identifying UFU vulnerabilities in Node.js? Can we simply apply existing solutions?
- What are the criteria to select the Node.js libraries and applications?

**Reviewer Confidence:**

3: The reviewer is confident but not certain that the evaluation is correct

**Scope:**

3: The work is somewhat relevant to the Web and to the track, and is of narrow interest to a sub-community

---

### Official Review · Reviewer_6LfZ · 2023-11-24

**Novelty:** 2
**Technical Quality:** 4

**Review:**

The paper describes presents a tool, named NODESEC, to analyze file upload vulnerabilities in nodeJS libraries and applications. Based on known UFU vulnerabilities enumerated from public sources like CVEs, the tool automates the process of analyzing the presence of these vulnerabilities. NODESEC is used to perform testing of popular nodsJS libraries and applications that results in identification of multiple UFU vulnerabilities in the existing applications.

Pros:
- Good evaluation results resulting in multiple CVEs and US-CERT cases
- NODESEC is open-sourced
- Responsible disclosures

Cons:
- Limited research contributions and novelty
- Impact of the UFU vulnerability is limited, so results are not entirely surprising

The NODESEC tool is a good engineering effort at best. Given that the implemented vulnerabilities are well known, it is basically a testing tool of existing vulnerabilities with known attack vectors. As a result, there is limited research novelty in the proposed work. While the results are insightful showing weaknesses in popular nodsJS applications, they are effectively a result of basic tests against known vulnerabilities.

Another major weakness of the paper is that it does not highlight the overall security impact of the UFU vulnerabilities themselves. It limits its analysis to just the ability to upload a file, however, without being able to execute the file the attack is quite limited. The paper provides no evidence of a real-world known attack that has been executed and if it is easy to execute such an attack. Moreover, enterprises often use other file scanning tools to label and filter files periodically. Without a ability to execute a complete attack, it is not entirely surprising that the current libraries and applications have not made it a high priority to fix the issues even when such UFU vulnerabilities have been know from prior CVEs. The paper should provide empirical or experimental evidence of the cases how uploaded files can be used to execute an attack.

The paper does not discuss any generic countermeasures that should be implemented to address UFU attacks. Is it is just a matter of fixing the implementation or a generic solution can be implemented?

While the libraries were selected based on popularity, how were the applications selected for evaluation?

Post Rebuttal:
The rebuttal does not address my major concern of research novelty -- it is an engineering tool at best. While having multiple CVEs is a positive result, I am still not convinced that it can result in a real-world attack. That was my second concern -- it needs additional effort from the attacker to somehow execute the file that has been uploaded. It is not clear how it can be done in the presence of other defense mechanisms. Sure, an attack can be demonstrated (thanks for the video), however, it does not make the attack any more practical.

**Questions:**

- What are the unique novel contributions of the work if the underlying attacks are already well known?
- Even if the attacker succeeds in uploading a file, how can a successful attack be carried out? Please provide evidence that such attacks have been carried out in real world or experimentally.

**Reviewer Confidence:**

4: The reviewer is certain that the evaluation is correct and very familiar with the relevant literature

**Scope:**

3: The work is somewhat relevant to the Web and to the track, and is of narrow interest to a sub-community

---

### Decision · Program_Chairs · 2024-01-22

**Decision:**

Accept

**Comment:**

The reviewers had very similar views of this paper: they appreciated the practical impact of this research and the engineering effort required to port existing ideas for UFU detection to the Node.js setting. On the other hand, reviewers criticized the lack of novelty with respect to the methodological aspects of this research. Since the paper presents a useful tool and the results of the empirical evaluation are convincing, I recommend this paper to accepted for a poster presentation.

 ---